# A Neurosurgical Perspective on Brain Metastases from Renal Cell Carcinoma: Multi-Institutional, Retrospective Analysis

**DOI:** 10.3390/biomedicines11092485

**Published:** 2023-09-07

**Authors:** Liliana Eleonora Semenescu, Ligia Gabriela Tataranu, Anica Dricu, Gheorghe Vasile Ciubotaru, Mugurel Petrinel Radoi, Silvia Mara Baez Rodriguez, Amira Kamel

**Affiliations:** 1Department of Biochemistry, Faculty of Medicine, University of Medicine and Pharmacy of Craiova, Str. Petru Rares nr. 2–4, 710204 Craiova, Romania; lilisicoe@yahoo.com (L.E.S.); anica.dricu@umfcv.ro (A.D.); 2Neurosurgical Department, Clinical Emergency Hospital “Bagdasar-Arseni”, Soseaua Berceni 12, 041915 Bucharest, Romania; dr_vghciubotaru@yahoo.com (G.V.C.); mara.silvia@icloud.com (S.M.B.R.); kamel.amyra@yahoo.com (A.K.); 3Department of Neurosurgery, Faculty of Medicine, University of Medicine and Pharmacy “Carol Davila”, 020022 Bucharest, Romania; 4Neurosurgical Department, National Institute of Neurology and Neurovascular Diseases, Soseaua Berceni 10, 041914 Bucharest, Romania; petrinel.radoi@umfcd.ro

**Keywords:** brain metastases, kidney cancer, renal cell carcinoma, neurosurgery, targeted therapy

## Abstract

Background: While acknowledging the generally poor prognostic features of brain metastases from renal cell carcinoma (BM RCC), it is important to be aware of the fact that neurosurgery still plays a vital role in managing this disease, even though we have entered an era of targeted therapies. Notwithstanding their initial high effectiveness, these agents often fail, as tumors develop resistance or relapse. Methods: The authors of this study aimed to evaluate patients presenting with BM RCC and their outcomes after being treated in the Neurosurgical Department of Clinical Emergency Hospital “Bagdasar-Arseni”, and the Neurosurgical Department of the National Institute of Neurology and Neurovascular Diseases, Bucharest, Romania. The study is based on a thorough appraisal of the patient’s demographic and clinicopathological data and is focused on the strategic role of neurosurgery in BM RCC. Results: A total of 24 patients were identified with BM RCC, of whom 91.6% had clear-cell RCC (ccRCC) and 37.5% had a prior nephrectomy. Only 29.1% of patients harbored extracranial metastases, while 83.3% had a single BM RCC. A total of 29.1% of patients were given systemic therapy. Neurosurgical resection of the BM was performed in 23 out of 24 patients. Survival rates were prolonged in patients who underwent nephrectomy, in patients who received systemic therapy, and in patients with a single BM RCC. Furthermore, higher levels of hemoglobin were associated in our study with a higher number of BMs. Conclusion: Neurosurgery is still a cornerstone in the treatment of symptomatic BM RCC. Among the numerous advantages of neurosurgical intervention, the most important is represented by the quick reversal of neurological manifestations, which in most cases can be life-saving.

## 1. Introduction

### 1.1. A New Epidemiological Threat?

With over 15 histological subtypes identified by the World Health Organization, renal cell carcinoma (RCC) comprises 3.8% of all new cancer diagnoses. A surge in the number of new cases has been reported, with more than 70,000 new cases of RCC in 2020 and more than 14,000 deaths in the last decade in the United States [1,2]. In Europe, in 2019, the European Association of Urology (EAU) concluded that the peak incidence of RCC occurs in Western countries, leading to approximately 100,000 new RCC cases and causing more than 39,000 deaths. In some European countries, mortality rates show a rising trend, while the annual increment worldwide during the last decades exceeded 2%, with a male predominance. The onset age is less variable, remaining at 60–70 years of age [1,2,3]. In Romania, in 2018, the prevalence of RCC over the last 5 years was approximately 5400 cases, with an annual incidence of 2000 cases, while in 2020, the prevalence was 7510, with 2750 new cases [4,5].

Regarding mRCC, it has been reported to occur in approximately 25–30% of patients with RCC, while BM from RCC have a described incidence of almost 13% [6]. The incidence of BM in individuals with ccRCC has been estimated to be 8% [7], with reported cases of leptomeningeal metastases [8]. In patients with non-clear cell RCC (nccRCC), brain involvement has been reported to be 3% in papillary RCC (pRCC) and 2% in chromophobic RCC (chRCC) [7]. Multiple BM RCCs have been reported in up to 45% of patients with mRCC [9].

It is important to specify that the increasing reported number of RCC cases is also due to advancements in imaging techniques. Concerning this subject, the state of the art is represented by the use of radiomics, which is a field that combines artificial intelligence, computer science, and radiology, in order to amplify the accuracy of medical imaging [9]. In spite of various challenges, this field has demonstrated great potential for diagnostic and prognostic purposes [10]. In the current management of RCC, radiomics can distinguish between RCC and angiomyolipoma, oncocytoma, and various subtypes of RCC, as well as preoperatively predict the nuclear grade and assess the therapeutic response [11]. In mRCC patients treated with Sunitinib, diffusion-weighted imaging (DWI)-MRI and positron emission tomography/MRI radiomics analysis were used as biomarkers in order to assess treatment response [12,13]. Lately, molecular imaging has also been of interest in RCC, as it also helps differentiate between distinct subtypes [11,14]. In addition, radiogenomics, which combines radiomics features with gene expression [15], has been proven to be of great help in patients with RCC, as it can predict therapeutical responses and prognosis [16,17,18].

### 1.2. Prognosis

The prognostic role of RCC is decisive in determining therapeutic management. For metastatic RCC (mRCC), prognostic information is given by anatomical, histological, clinical, and molecular factors. The most important anatomical prognostic factor (PF) is represented by the tumor-node-metastasis (TNM) stage classification, but when it comes to clinical PFs, Karnofsky performance status (KPS) has persistently been the paramount determinant of survival. Regarding the histological PFs, RCC subtype, microvascular and collecting system invasion, Fuhrman nuclear grade, tumor necrosis, and sarcomatoid features are the most relevant. As for molecular PFs, worth mentioning are the expression of the Ubiquitin Carboxy-Terminal Hydrolase (BAP1) and Polybromo 1 (PBRM1) genes, chromosomal losses of regions 9p, 9q, and 14q, and CpG methylation-based assays [19,20,21].

### 1.3. Molecular Pathogenesis

The complexity of brain metastases from renal cell carcinoma (BM RCC) needs an understanding of this extremely heterogeneous pathology on a molecular level in an effort to address the poor prognosis. In order to succeed, treatments require adaptation to molecular susceptibilities among primary cancers and their metastases [22]. Renal cell carcinoma (RCC) is comprised of several subtypes with genetic drivers, individual histology, clinical evolution, and therapeutic responses. According to The Cancer Genome Atlas analyses of RCC, there are cardinal dissimilarities between the major histological subtypes, including metabolic pathway expression signatures and their distinct chromosomal alterations [23]. The most common subtypes of RCC are represented by the ccRCC, accounting for ~75% of cases, followed by papillary RCC (pRCC, type 1—basophilic and type 2—eosinophilic), and the chromophobic subtype (chRCC), which accounts for 5–15% of cases. The most investigated subtype is represented by the ccRCC, given the fact that it is the most frequent. The chromosome that has known relations with ccRCC is 3p. Genetic mutations in ccRCC are described in the following genes: the von Hippel–Lindau tumor suppressor gene (VHL, most frequent, up to 50%), PBMR1 (up to 40%), BAP1 (up to 15%), and other genes recounted as SET Domain Containing 2 (SET D2), Phosphatidylinositol-4,5-Bisphosphate 3-Kinase Catalytic Subunit Alpha (PIK3CA), and also TSC Complex Subunit 1 and 2 (TSC1/2) [23,24,25,26]. The last two genes are responsible for the activation and suppression of mammalian target of rapamycin (mTOR) pathways and have been defined as being involved in metabolic RCCs. The first two mentioned genes manage the regulation of hypoxia-inducible factor (HIF) protein and the encoding of BAF180 (equivalent name of PBMR1), respectively. Although the VHL gene is described in more than half of the patients with this pathology, other genetic alterations may arise progressively, worsening the prognosis. Regarding the pRCC, genetic alterations are described in the following genes: Mesenchymal Epithelial Transition (MET), Cyclin-Dependent Kinase Inhibitor 2A/B (CDKN2A/B), Telomerase reverse transcriptase (TERT), and Fumarate Hydratase (FH), the first one being the most frequent. Additionally, in the sarcomatoid subtype, mutations were described in Tumor Protein P53 (TP53), a tumor suppressor inscribing for p53, and Moesin-Ezrin-Radixin Like (MERLIN) Tumor Suppressor (NF2) [3,19,27,28].

### 1.4. Therapeutic Approaches in BM RCC

#### 1.4.1. Hereditary versus Sporadic, and the Primary Management of RCC

The CT scan may be extremely helpful in the evaluation of RCC, with greater than 95% accuracy. A contrast-enhanced CT scan reveals an enhanced renal mass, which is a solid clue for the diagnosis of the disease. While searching for diagnostic certainty, the histopathological examination after a tissue biopsy may be essential in the diagnosis and management of RCC [29]. Subsequently, similar to CT scans, MRI and ultrasonography can be utilized to describe a specific stage before any treatment is given. Hereditary proneness to RCC is guided by the existence of many factors, of which the following are described: age younger than 50 years old, multiple enhancing lesions, and/or a family history of RCC. A thorough physical examination is required in order to assess extrarenal manifestations (e.g., ophthalmologic, neurologic, and dermatologic evaluations) [30,31,32]. For instance, compared to Birt–Hogg–Dubé syndrome, von Hippel–Lindau disease, and RCC, which may cause systemic damage, family ccRCC and hereditary pRCC do not reveal systemic manifestations other than renal. Genetic testing can be used to identify mutations in specific genes, as it was concluded that the von Hippel–Lindau protein has a major role in sporadic ccRCC. Monitoring and evaluation form the basis for patients with hereditary RCC to assess new renal masses, and thus, imaging studies must be carried out, sometimes with an interval difference of 6 months or more, determined by the character of the syndrome and the lesions. Finally, in the multimodal therapeutic approach, a key element is defined by the surgical treatment in selected cases (i.e., radical nephrectomy, nephron-sparing partial nephrectomy or laparoscopic nephrectomy, surgery for metastatic disease, percutaneous ablative approaches) and targeted therapy [23,32,33,34].

#### 1.4.2. Multidisciplinary Opportunities in BM RCC

RCC is well known for the low rates of response to chemotherapy and radiotherapy, which gave rise to much research regarding the development and evolution of targeted therapies, including vascular endothelial growth factor (VEGF) and transforming growth factor alpha (TGF-α) pathways, mTOR inhibitors, immune checkpoint inhibitors (ICI), and more recently, combined targeted therapies. Anti-angiogenic strategies became alluring given the fact that RCC is a decidedly vascular cancer. Despite their high initial effectiveness, these agents often fail as tumors become resistant or relapse, and therefore some patients experience disease progression. Afterward, in advanced RCC, patients will frequently develop BM, and more than 80% are symptomatic at the moment of diagnosis [35,36,37,38,39].

Hence, despite the increased availability of targeted agents along with multimodal therapies, neurosurgery is still the main pawn when approaching BM RCC in symptomatic patients, especially with solitary BM. Among the numerous advantages of surgical intervention, the most important is represented by the quick reversal of neurological manifestations [40,41]. During neurosurgical resection, tumor tissue may be obtained for histopathological examination in favor of genetic tumor characterization. Notwithstanding its advantages, the neurosurgical approach alone is insufficient for the local control of the tumor, and the craniotomy is not risk-free, carrying a mortality rate of less than 2% and a morbidity rate between 4 and 6% [42,43]. However, accompanied by other therapies, neurosurgical treatment was proven to extend the overall survival (OS) rate.

Although RCC is appraised to be radioresistant, some studies confirm the opposite, stating that in a multimodal approach, stereotactic radiosurgery (SRS) might be a major factor in the management of the disease [44]. With the advent of minimally invasive techniques, it is now possible to decrease the disclosure of the cerebral tissue and the discomfort, maximize the safety of the central nervous system approach, reduce recovery time, and therefore increase the advantages of the method [44,45]. Some of these techniques are represented by stereotactic laser ablation (which requires the insertion of a laser catheter through the burr hole), convection-enhanced delivery (also a form of intratumoral therapy), focused ultrasound (for gaining access deep within the brain to ablate only the target tissue without harming the surrounding tissue), stereotactic laser interstitial thermal therapy (LITT, a cytoreductive neurosurgical technique), and stereotactic biopsy (most common when there is more than one lesion and with deep localization) [23,46,47,48].

An important part of the multimodal therapeutic approach, which can improve overall survival rates, includes cytoreductive nephrectomy (CN) [49]. CN is defined as the removal of the primary RCC tumor lesion in the presence of metastases [50]. Regarding the numerous benefits of CN, it is worth mentioning that it improves the quality of life by alleviating symptoms (e.g., hematuria, pain) [50] and removes a potential source for new metastases [51,52,53]. However, CN has disadvantages as well, including perioperative morbidity and mortality, as well as deferred receiving of systemic therapy [52]. Furthermore, studies like CARMENA and SURTIME highlight the paramount aspect of careful patient selection, as not every patient may benefit from CN [54,55]. Notwithstanding, patients with favorable-risk features are more likely to benefit from CN [56,57], although approximately 20% of patients with nephrectomy will still develop metastases [58]. Despite numerous studies and debates, CN remains a moving target surrounded by controversy, highlighting the need for future studies on the matter [50,59,60].

#### 1.4.3. A Focus on Neurosurgery

The colonization of metastases in the brain is a result of tumor cells spreading throughout the blood, as well as seeding from an already existing metastasis in the body [22]. Microenvironmental interactivity, neuroinflammatory cascades, and neovascularization are the basis of developing BM [22]. Concerning the BM RCC, it is well known that this pathology has a greater tendency for vasogenic edema and hemorrhage; therefore, patients are oftentimes symptomatic [61]. Nevertheless, neuroimaging is usually recommended in symptomatic patients or at the doctor’s discretion [36,61], as well as if clinically indicated [62]. However, currently, the National Comprehensive Cancer Network recommends routine neuroimaging in patients with mRCC, which is rather helpful in detecting asymptomatic BM [61]. Patients who, at the initial diagnosis, have BM usually exhibit a poor prognosis and, when left untreated, may have a median OS to the utmost of 4 months [6,63].

It is worth mentioning that several therapies can mimic intracranial disease progression, while new or incremented neuroimaging abnormalities in the course of immunotherapy or SRS may constitute pseudoprogression [64].

At present, the primary approach to BM RCC comprises neurosurgery and/or radiotherapy (RT) [65,66,67].

The selection of the patients is of great importance, as the decision of neurosurgical intervention must consider the advantages and disadvantages when compared to other therapeutic options. At the present time, neurosurgical resection can be considered a safe option, as it is correlated with minimal morbidity and mortality [68,69]. Nonetheless, when considering neurosurgery, several factors must be taken into account, like the status of primary cancer, Karnofsky performance status (KPS), the localization of BM in the brain areas, and patient characteristics [70]. The type of neurosurgical excision has a great impact on the clinical outcomes as well. En bloc resection and SRS are associated with better outcomes when compared to piecemeal resection, as the latter carries a higher risk of leptomeningeal dissemination in patients with single supratentorial BM [71]. In like manner, individuals with a single BM treated with neurosurgery and RT have better survival rates and quality of life in comparison to RT alone [72]. However, the therapeutic approach for patients with single BM RCC differs from patients with multiple BM RCC.

In patients with single BM, if asymptomatic and smaller than 3 cm, or if not fit for neurosurgery, SRS alone or fractioned stereotactic RT (FSRT) might be the option [73]. Individuals with lesions of 3 cm or bigger, especially if symptomatic, are candidates for neurosurgical excision. If patients are unfit for craniotomies, FSRT is preferred over single-fraction SRS [74]. It should be noted that despite general knowledge regarding the resistance of BM RCC to RT, various studies have proven otherwise [74,75]. Laser interstitial thermal therapy/ablation (LITT) might be considered in patients unfit for both neurosurgical resection and SRS [76], as in recurrent patients after SRS, LITT had similar outcomes to craniotomy [77].

In patients with multiple BMs, aggressive treatment for intracranial disease in oligometastatic cases has better outcomes when compared to whole-brain RT (WBRT) [78,79]. It has been concluded that WBRT has limited recommendations given its known relative resistance in RCC [70].

Given the fact that neurosurgery has a main role in BM RCC, our current study focuses on this exact matter in order to conclude to what extent patients with this pathology may benefit from it.

## 2. Materials and Methods

The authors of this study aimed to evaluate patients presenting with BM from RCC and their outcomes after being treated in the Neurosurgical Department of Clinical Emergency Hospital “Bagdasar-Arseni”, Bucharest, Romania, and the Neurosurgical Department of the National Institute of Neurology and Neurovascular Diseases, Bucharest, Romania. We included adult patients with renal cell carcinoma as their only malignancy, and we excluded patients with more than one malignancy and patients who underwent prior neurosurgical interventions in other neurosurgical departments. In order to make the selection, registry databases from our departments, as well as the patient’s physical file, were queried for all patients with histologically confirmed BM from RCC from 2012 to 2022 and evaluated retrospectively. The selected data to review were represented by the clinical notes, demographics, histology, comorbidities, BM topography, neurosurgical treatment, systemic therapy, extracranial metastases, prior nephrectomy, and outcomes. Regarding the matter of cytoreductive nephrectomy, when admitted to our departments, 37.5% of patients with BM RCC had already undergone surgery for the primary RCC lesion. Of the total of 37.5% (9 patients), 66.6% were from urban settings.

The current study is based on a thorough appraisal of the patient’s demographic and clinicopathological data and is focused on the strategic role of neurosurgery in the multimodal therapy of BM from RCC.

## 3. Results

### 3.1. Statistics and Replicability

This retrospective study was carried out by reviewing the medical records from our institutional databases of 24 patients treated in our departments between January 2012 and December 2022 for BM RCC. We appraised demographic information, clinicopathological characteristics, as well as therapeutic options. Correlations between the obtained data were performed in order to draw conclusions. Statistical analyses of experimental data, figures, and tables were performed using GraphPad Prism 8.3.0 software. *p*-values less than 0.05 were considered statistically significant. The one-sample *t*-test, one-way/two-way ANOVA, and the Chi-square (and Fisher’s exact) were used to assess differences in variables and correlate normally distributed data. Kaplan–Meier method was the choice for performing survival analysis. Overall survival was interpreted as the period between the histopathological diagnosis and the date of death.

### 3.2. Demographic Profile, Clinicopathological Characteristics, and Correlation Analysis

Within our institutions, 24 patients (n = 24) were admitted with BM RCC (Table 1), of whom 20.8% were women and 79.1% were men. Ten (41.6%) patients lived in urban areas, while 14 (58.3%) were living in rural settings. The median age of BM RCC diagnosis was 62.5 years (36–73), while the mean age of RCC diagnosis was 62 years. While a total of 9 (37.5%) patients underwent nephrectomy, regarding the comorbidities, 11 patients (45.8%) had a history of congestive heart failure and heart disease, and 7 (29.1%) had a history of type 2 diabetes mellitus. Seven (29.1%) patients received systemic therapy, and only two (8.3%) patients were asymptomatic at admission. The most common clinical symptoms were represented by headache (45.8%, n = 11) and limb paralysis (41.6%, n = 10), while aphasia (12.5%, n = 3), seizures (8.3%, n = 2), and pituitary dysfunction (4.1%, n = 1) were less frequent. Regarding the localization of BM RCC, the most frequent site was represented by the frontal lobe (33.3%, n = 8), followed by the cerebellum (29.1%, n = 7), and the temporal lobe (25%, n = 6). The least common sites of metastasis were represented by the parietal lobe (4.1%, n = 1), the occipital lobe (4.1%, n = 1), and the sellar region (4.1%, n = 1) (Figure 1). When comparing the distribution of lesion size by symptom status, we obtained a statistically significant result (*p* = 0.042), showing that patients with smaller lesions were more likely to be symptomatic (Figure 2). These results may be due to the localization of BM in neurologic eloquent areas.

Although Romania is still a developing country with a high-income economy, in rural areas there is limited availability of healthcare resources. In our study, 9 (37.5%) patients underwent nephrectomy. From the total number (n = 9), 66.6% were from urban settings in comparison with half of that percentage (33.3%) that were from rural areas. In the group without nephrectomy (n = 15), 11 (73.3%) were from rural areas and 4 (26.6%) from urban settings; *p* = 0.058 (Figure 3).

In our study population, lesion sizes between 13 and 30 mm had a homogeneous distribution in both rural and urban settings (25% versus 25%), while the number of patients with lesion sizes between 30 and 63 mm doubled in rural areas: 33.3% in rural areas versus 16.6% in urban settings (Figure 4). However, the conclusion was not statistically significant; *p* = 0.375.

### 3.3. Neurosurgical Results

Out of 24 patients with BM RCC, in 23 cases a neurosurgical resection was performed, and in 1 case SRS alone was indicated. The main purposes of neurosurgical intervention were gross-total resection (which occurred in 18 patients) and relief of mass effects in order to improve neurological symptoms. Due to the infiltration and adherence of BMs, as well as their localization in eloquent brain areas, only partial resection was possible in five patients (Figure 5). In cases with multiple BMs, we approached only the symptomatic lesion.

Regarding the post-surgical complications, a postoperative intracerebral hematoma has been noted in one patient, and it was operated on without further consequences. The initial KPS in patients with BM RCC was <80 in 11 (45.8%) cases (Figure 6). After neurosurgery, the score improved in 15 (62.5%) cases, remained unchanged in 8 (33.3%) cases, and worsened in 1 (4.1%) case.

Seven (29.1%) of our patients had intracranial metastases: six cases of lung metastases and one case of intestinal metastasis. The remaining 17 individuals did not have extracranial metastases.

Three (12.5%) patients in our study population were treated with SRS for BM RCC. One patient with a small BM (13 mm) was treated with SRS alone, while two patients were treated with SRS and neurosurgical resection.

### 3.4. Histopathological Features

The final diagnosis based on histopathology of the neurosurgical specimen was ccRCC in 91.6% of the patients (Figure 7) and pRCC in 8.3% of the patients.

### 3.5. Survival Analysis

The probability of survival was higher in patients who underwent nephrectomy than in those who did not. Overall survival was also extended in those who underwent nephrectomy (log-rank [Mantel–Cox] test). The median survival rate in patients with nephrectomy was 12 months (hazard ratio [HR] = 0.36, 95% CI 0.16–0.83), while in patients without nephrectomy, it was 7 months (HR = 2.72, 95% CI 1.19–6.21); *p* = 0.004 (Figure 8).

The probability of survival was higher for patients treated with systemic therapy. The median survival rate in patients treated with systemic therapy was 13 months (HR = 0.41; 95% CI 0.18–0.91), while in patients without systemic therapy it was 12 months (HR = 2.42; 95% CI 1.09–5.40), *p* = 0.033 (Figure 9).

In the study population, 4 (16.6%) patients had multiple BM RCC, and 20 (83.3%) had a single BM. Compared to patients with more than one BM, those with a single BM were observed to have better survival rates: Median survival in patients with one BM was 9.5 months (HR = 2.11, 95% CI 0.72–6.17) vs. 4.5 months (HR = 0.47, 95% CI 0.16–1.38) in patients with more than one BM; *p* = 0.0005 (Figure 10).

In order to provide the best therapeutic options and to predict survival rates, patients with RCC were classified according to the International Metastatic Renal Cell Carcinoma Database Consortium (IMDC) risk subgroup, which currently represents the gold standard [63,64] (Table 2).

In our study, the majority of the population was in the intermediate IMDC risk subgroup (66.6%). We calculated the survival rates in each of the three subgroups (Figure 11). The favorable-risk subgroup had the longest median survival, 13 months, followed by the intermediate-risk subgroup, with a 9-month median survival. The shortest median survival rate has been registered in the poor-risk subgroup, with a median survival rate of 4 months; *p* < 0.0001.

### 3.6. The Metabolism of Ferrous Iron: A Soft Spot in the Modulation of Cancer and Metastasis?

Back in 2005, Herbert T. Cohen and Francis J. McGovern stated in an article regarding medical progress in RCC that, among others, a low hemoglobin level predicts a poor prognosis [80]. However, in 2022, Honglin Jiang et al. found that oncogenic KRAS signaling induced ferrous iron accumulation and that elevated iron concentrations in some types of cancer are correlated with a lower survival rate. They were looking forward to exploring their ferrous iron-activatable drug conjugate (FeADC) technology, which is in effect a converted FDA-approved MEK inhibitor, in order to achieve MAPK blockade in cancerous cells [81]. However, Richard E. Gray et al. stated in an article about the diagnosis and management of RCC that a major indicator of poor prognosis is, among others, a low hemoglobin level [35,82,83]. In our study, the median value of hemoglobin was 13.7 g/dL. The normal levels of hemoglobin established by our laboratory were 12–15 g/dL for female patients and 13–17 g/dL for male patients; levels below were considered indicators of anemia, while levels above were considered elevated. The lowest value for hemoglobin in the study group was 8.8 g/dL, while the highest was 18.4 g/dL (Figure 12a). Anemia was described in 20.8% of the male population, 8.3% of the female population, and 29.1% of the total. However, higher levels of hemoglobin were associated in our study with a higher number of BM, and it was statistically significant; *p* < 0.001 (Figure 12b).

The median survival in patients with normal levels of hemoglobin was 9 months, which was the same as in low hemoglobin cases, while individuals with elevated levels of hemoglobin had a lower median survival rate (median survival of 8 months; Figure 13). However, the comparison was not statistically significant (*p* = 0.583).

## 4. Discussion

Despite the newly increased availability of multimodal therapies, BM RCC have generally poor prognostic features with dismal outcomes [69]. Notwithstanding these classically considered results, our study group experienced durable long-term survival, as patients appear to benefit from the multimodal therapeutic approach. The longest survival rate, of 19 months, has been achieved in a 63-year-old patient from a rural area with a single 24 mm cerebellar BM RCC who was admitted for cerebellar syndrome. The patient also had a pulmonary metastasis, underwent nephrectomy before neurosurgery, and did not receive systemic therapy. The shortest survival rate, of 3 months, was registered in two patients aged 70 and 73, respectively. One of them received systemic therapy, while the other did not. 

Although not every patient with RCC will benefit from nephrectomy, some individuals will experience OS benefits, especially in the context of immune therapy [70,71,72]. In our study, 37.5% of patients with BM RCC had a prior nephrectomy (before being admitted to our departments). In this group, longer survival rates were registered, with a median survival rate of 12 months (HR = 0.36, 95% CI 0.16–0.83), in comparison to 7 months (HR = 2.72, 95% CI 1.19–6.21) in patients without nephrectomy (*p* = 0.004). When comparing this treatment with the geographic region of origin of our patients, we found that 73.3% of the patients without nephrectomy were from rural areas in comparison to only 26.6% from urban settings (*p* = 0.058). This statement is specifically significant because in Romania, there is limited availability of healthcare resources in most of the rural areas. Similarly, regarding the multimodal approach of BM RCC, in our study group, a total of 7 (29.1%) patients received systemic therapy in comparison to 17 (70.8%) who did not. Our study demonstrates that the median survival rate in patients treated with systemic therapy was longer than in those without systemic therapy: 13 months versus 12 months.

Likewise, when focusing on the number of BM RCC at the diagnosis, compared to patients with more than one BM, those with a single BM were observed to have better survival rates. Median survival in patients with one BM was 9.5 months (HR = 2.11, 95% CI 0.72–6.17), in comparison to 4.5 months (HR = 0.47, 95% CI 0.16–1.38) in patients with more than one BM; *p* = 0.0005. It is worth mentioning that only four patients from the study group had multiple metastases, while the rest of the population had a single BM RCC. Four patients (16.6%) experienced local recurrence, of whom one had a single BM RCC, while the other three patients had two or more than two BM RCC.

When comparing the distribution of lesion size by symptom status, we found that patients with smaller lesions were more likely to be symptomatic (*p* = 0.042). Nine symptomatic patients (37.5%) had lesions of sizes between 13 and 27 mm, eight symptomatic patients (33.3%) had lesions of sizes between 30 and 40 mm, three symptomatic patients (12.5%) had lesions between 43 and 52 mm, and two symptomatic patients (8.3%) had lesions between 52 and 63 mm. One asymptomatic patient (4.1%) fell into the category of the 13–27 mm lesion size group, and one asymptomatic patient fell into the category of the 30–40 mm lesion size group. A possible explanation of this result could be the fact that the localization of the tumors, regardless of their small size, was in areas of the brain with important neurological and/or neuroendocrinological functions (i.e., frontal lobe, temporal lobe, pituitary region). Even if it was not statistically significant, it is worth mentioning that in our study group, the number of patients with lesion sizes between 30 and 63 mm has doubled in rural areas (33.3% in rural areas versus 16.6% in urban settings, *p* = 0.375). 

In our study population, in 23 patients, a neurosurgical resection of BM RCC was performed, with 18 gross-total resections and 5 partial resections. The partial resections were justified by tumoral infiltration and adherence, as well as localization in eloquent brain areas. One patient had a postoperative intracerebral hematoma. An emergency surgery was performed for evacuation, and the postoperative outcome was not affected by this event. No other postoperative complications have been noted.

KPS at admission in our departments was lower than 80 in 11 (45.8%) cases and higher in 13 (54.1%). After the neurosurgical intervention, the score improved in 15 (62.5%) cases, which represents more than half of the patients. 

The mainstay of treatment for symptomatic patients with BM RCC is still the classical neurosurgical approach, which first and foremost offers a quick reversal of neurological manifestations. However, neurosurgery alone is insufficient, but along with other therapies, it offers long-lasting local control and may extend the OS rate [41,84]. 

Even though RCC is considered radioresistant, it has been proven that SRS may improve the initial poor prognosis in a multimodal approach [45,85,86,87]. However, unfortunately, our study consisted of only three patients who underwent SRS, so we consider this one of the study’s limitations. Only one patient was treated by SRS alone, while the other two were treated with SRS and neurosurgical resection. Given the very small group of patients treated with SRS, we did not conclude a significant comparison between groups and did not include statistical analysis in this article. However, it is worth mentioning that one patient treated with SRS and neurosurgical resection had a survival of 19 months, the other one of 13 months, and the patient treated with SRS alone had a survival of 13 months. In the latter case, the patient had a single 13 mm BM, no comorbidities, minimal symptoms at a first-time admission, a prior nephrectomy, and no systemic therapy received.

In order to predict survival rates and provide the best therapeutic options for patients with RCC, the IMDC risk-scoring system has been designed [88,89]. In our study population, 5 patients were in the favorable-risk group, 16 in the intermediate-risk group, and 3 in the poor-risk group. When comparing survival rates, the first group had the longest median survival rate (13 months), followed by the intermediate-risk subgroup (9 months). The shortest survival was concluded in the poor-risk subgroup, and it was statistically significant (*p* < 0.0001). 

A recent study by Honglin Jiang et al. discovered that in some categories of cancer, elevated iron concentrations are correlated with a lower survival rate [81]. The authors stated that an increased level of Fe2+ is also linked to drug tolerance in cancer cells, but the mechanism is still undetermined. Moreover, the authors concluded that intracellular Fe2+ is elevated by mutant RAS signaling [81]. Even though these statements represented just the basis of the study, starting from this point and inspired by other previous studies [90,91,92,93], we sought to evaluate this perspective. In our study, normal levels of hemoglobin established by our laboratory were 12–15 g/dL for female patients and 13–17 g/dL for male patients, while the levels below were considered indicators of anemia. Twelve (50%) patients had modified hemoglobin levels, and 12 were in the normal range. The median hemoglobin level in the study group was 13.7 g/dL, with the highest level of 18.4 g/dL and the lowest of 8.8 g/dL. Anemia was described in 29.1% of the study population. Interestingly, it turned out that higher levels of hemoglobin were associated with a higher number of BM, and higher numbers of BM were associated with lower survival rates.

It has been known for a long time that some cases of RCC have a hereditary constituent, and the most common examples encompass von Hippel–Lindau syndrome and the familial pRCC syndrome [94,95]. However, in our series of BM RCCs, none of the patients have been diagnosed with hereditary RCCs.

Many of the patients in our study had comorbidities that may place them at a higher risk for chronic kidney disease or even end-stage renal disease. More precisely, as many as 45.8% of patients had a history of congestive heart failure or heart disease, and 29.1% had a history of diabetes. In one of their articles, Tyler Clemmensen et al. suggested that partial nephrectomy is especially important in this category of patients as a nephron-sparing approach [1].

Overall, consistent with prior studies, the authors of this article established that neurosurgical patients with BM RCC may benefit from multimodal therapeutic approaches. Notwithstanding that neurosurgery is the gold standard for symptomatic patients with BM RCC, it alone does not suffice and, in addition to other therapies, may increase survival rates. 

One of the main limitations of the current study is represented by the small sample size, which may have resulted in insufficient statistical power to permit the detection of significance for some variables (i.e., the Chi-square test is not accurate in small samples). Furthermore, the study was also limited by the small group of patients who were treated with radiation therapy (SRS). Among the limitations, we also consider the exclusion criteria regarding patients with BM RCC initially treated in other neurosurgical departments. In the current study, we solely included patients neurosurgically treated in our departments from the beginning for a better follow-up and evolution assessment. However, we are looking forward to including in future studies patients with possible recurrences initially treated in other neurosurgical departments.

To our knowledge, this is the first study focused on BM RCC patients in Romania that involves a multicentric approach, and for a better understanding of the topic, we encourage other studies regarding the matter. 

## 5. Conclusions

Although the expectancy of life’s duration is rather short in patients with BM RCC, neurosurgical approaches combined with other therapeutical options can offer long-lasting local control and can increase survival rates. Moreover, histopathological examination of tumoral tissue can be obtained in order to establish the best targeted therapeutic agents. Among the numerous advantages of neurosurgical intervention, the most important is represented by the quick reversal of neurological manifestations, which in most cases can be life-saving. Despite the increased availability of targeted agents along with multimodal therapies, neurosurgery is still a cornerstone in the treatment of BM RCC in symptomatic patients.

## Figures and Tables

**Figure 1 biomedicines-11-02485-f001:**
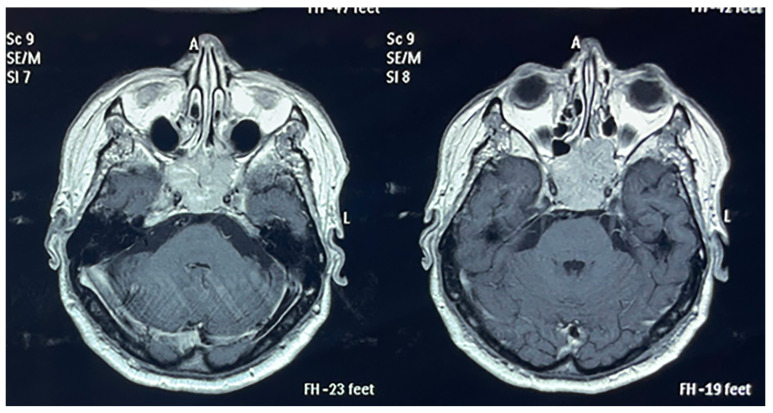
Pituitary MRI reveals a non-homogenous invasive sellar mass (BM) from RCC in a patient from our study group.

**Figure 2 biomedicines-11-02485-f002:**
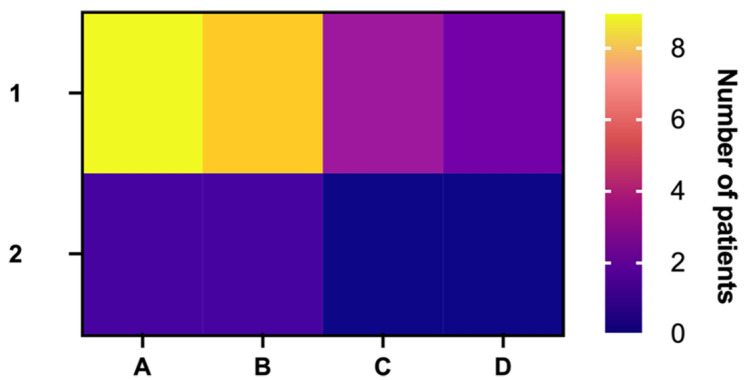
Heat map comparing the distribution of lesion size by symptom status. 1—symptomatic patients; 2—asymptomatic patients. Size of the lesion (mm): A—13–27; B—30–40; C—43–52; D—52–63.

**Figure 3 biomedicines-11-02485-f003:**
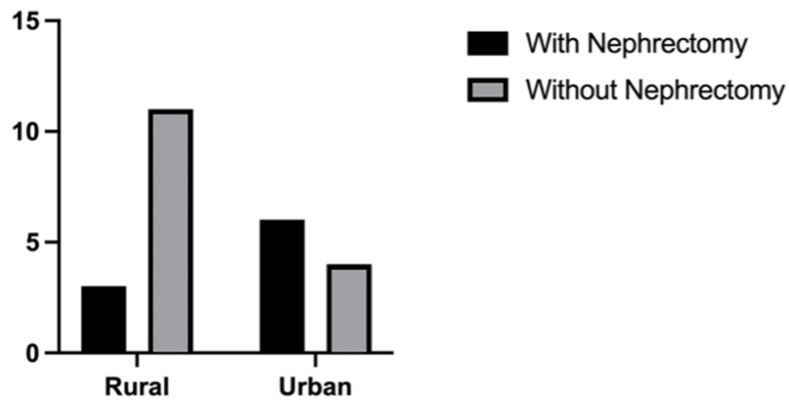
Comparison between patients from rural areas with and without nephrectomy versus patients from urban settings with and without nephrectomy (*p* = 0.058).

**Figure 4 biomedicines-11-02485-f004:**
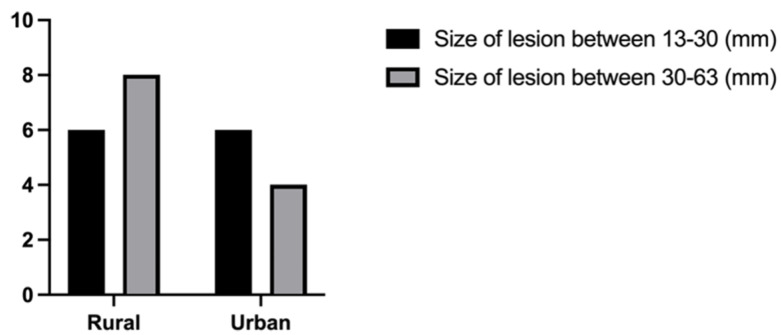
Bar graph comparing the distribution of lesion size by geographic areas (*p* = 0.375).

**Figure 5 biomedicines-11-02485-f005:**
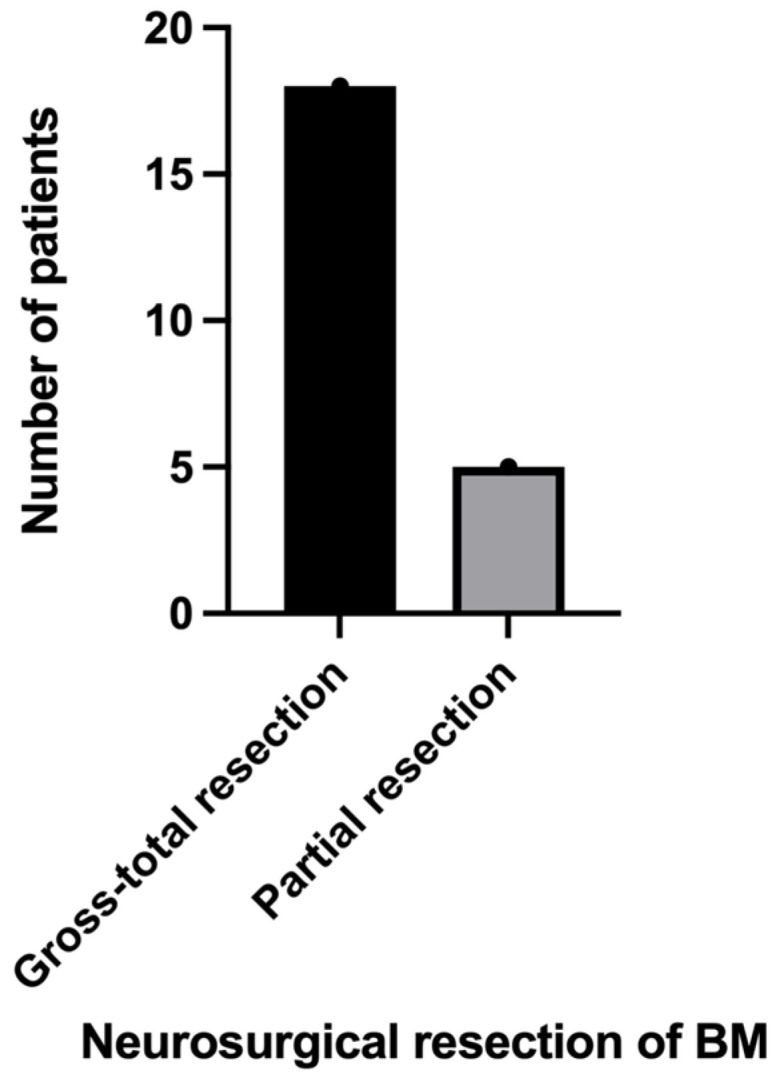
Bar graph describing the neurosurgical resection of BM RCC in our study population.

**Figure 6 biomedicines-11-02485-f006:**
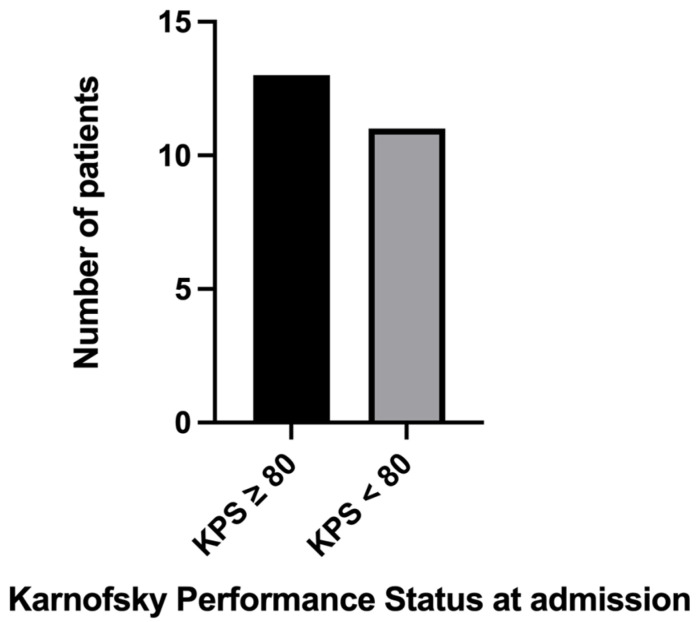
Bar graph describing KPS at admission in our departments, in patients with BM RCC.

**Figure 7 biomedicines-11-02485-f007:**
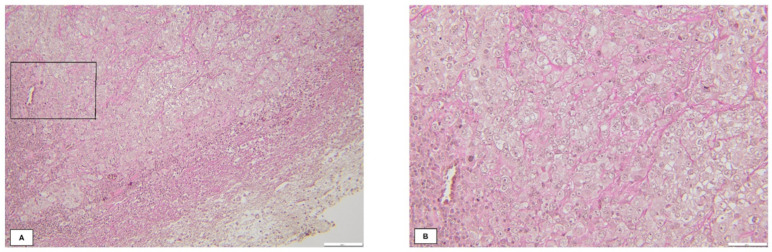
Photomicrograph of BM RCC exhibiting clear cell morphology and a prominent network of thin-walled vessels by Van Gieson staining. Magnification: (**A**)—20×, (**B**)—40×. Contributed by Laurențiu-Cătălin Cocoșilă, M.D.

**Figure 8 biomedicines-11-02485-f008:**
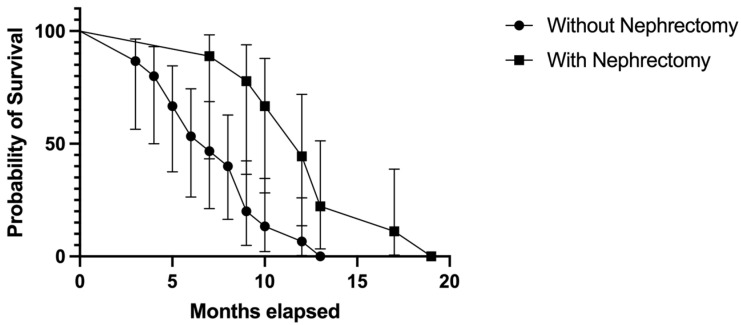
Kaplan–Meier plot describing the survival proportion in patients with and without nephrectomy.

**Figure 9 biomedicines-11-02485-f009:**
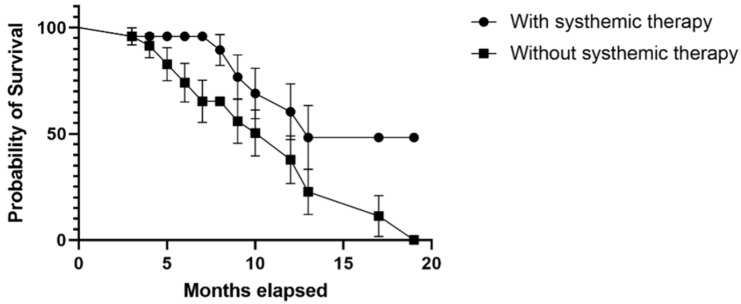
Kaplan–Meier plot describing the comparison of survival rates between neurosurgical patients treated with systemic therapy and patients who did not receive systemic therapy.

**Figure 10 biomedicines-11-02485-f010:**
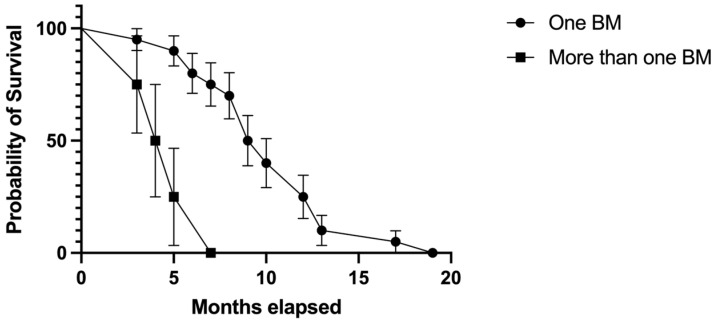
Kaplan–Meier plot describing the comparison between survival rates in patients with one BM RCC versus patients with two or more BM RCC.

**Figure 11 biomedicines-11-02485-f011:**
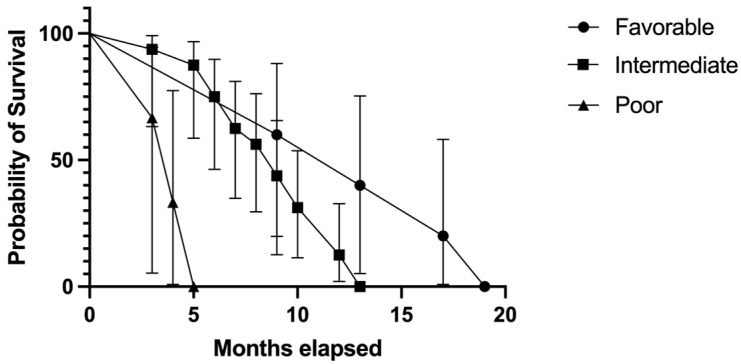
Kaplan–Meier plot describing the comparison between IMDC risk subgroups in patients with BM RCC in our study.

**Figure 12 biomedicines-11-02485-f012:**
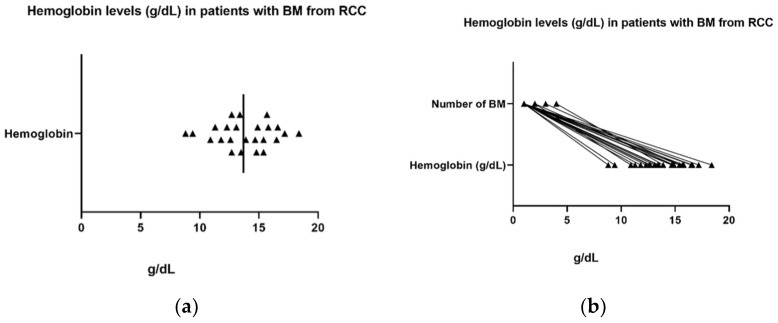
(**a**) Hemoglobin levels in patients with BM RCC in our study population. The median hemoglobin level was 13.7 g/dL, with the highest level of 18.4 g/dL and the lowest of 8.8 g/dL; (**b**) hemoglobin levels were associated with the number of BM in patients with RCC in our study population (*p* < 0.001; mean difference = 1.29, 95% CI 0.97–1.61).

**Figure 13 biomedicines-11-02485-f013:**
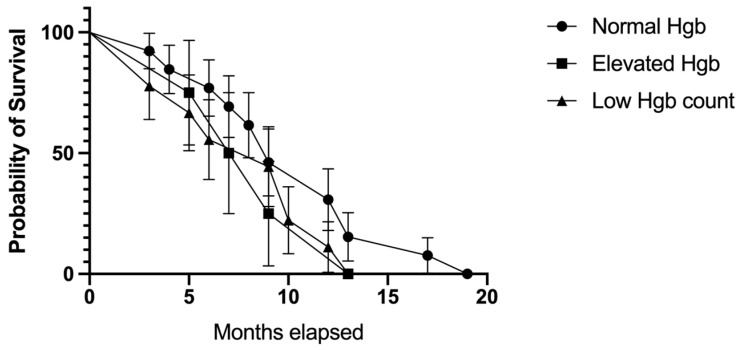
Kaplan–Meier plot describing the comparison between the survival rate in patients with normal, elevated, and low hemoglobin levels; *p* = 0.210.

**Table 1 biomedicines-11-02485-t001:** General characteristics of patients with BM from RCC in our study group.

Characteristic	Variable	Value (total n = 24)
	Male	19 (79.1%)
Sex	Female	5 (20.8%)
Distribution area	Rural/Urban	14 (58.3%)/10 (41.6%)
Age at RCC diagnosis	Median age	62 years
Age at BM RCC diagnosis	Median age	62.5 years
BM localization	Frontal lobe	8 (33.3%)
	Temporal lobe	6 (25%)
	Parietal lobe	1 (4.1%)
	Occipital lobe	1 (4.1%)
	Cerebellum	7 (29.1%)
	Sellar region	1 (4.1%)
Single or multiple BM RCC	Single BM RCC 2 or more BM RCC	20 (83.3%) 4 (16.6%)
Number of BM RCC	Mean/Median (min-max)	1.29/1 (1–4)
Size of BM RCC (mm)	Mean/Median (min-max)	32.54 mm/31 mm (13–63)
Symptoms at presentation	Yes No	22 (91.6%) 2 (8.3%)
Clinical symptoms/manifestations	None	2 (8.3%)
	Raised intracranial Pressure syndrome	7 (29.1%)
	Headache	11 (45.8%)
	Cranial nerve palsies	5 (20.8%)
	Pituitary dysfunction	1 (4.1%)
	Limb paralysis	10 (41.6%)
	Aphasia	3 (12.5%)
	Seizures	2 (8.3%)
Karnofsky Performance Status Scale at admission	≥80 <80	13 (54.1%) 11 (45.8%)
Admission to the neurosurgical department	First time Recurrence	20 (83.3%) 4 (16.6%)
Extracranial metastases ^1^	Yes No	7 (29.1%) 17 (70.8%)
	Yes	7 (29.1%)
Systemic therapy	No	17 (70.8%)
	Yes	9 (37.5%)
Prior nephrectomy	No	15 (62.5%)
	Yes	3 (12.5%)
Stereotactic Radiosurgery (SRS)	No	21 (87.5%)

^1^ 7 (29.1%) of our patients had extracranial metastases (lung and intestinal metastases).

**Table 2 biomedicines-11-02485-t002:** IMDC risk subgroups in our study population.

IMDC Risk	Patients N (%)
Favorable-risk group	5 (20.8%)
Intermediate-risk group	16 (66.6%)
Poor-risk group	3 (12.5%)

## Data Availability

Datasets analyzed or generated during the study are unavailable due to privacy and ethical restrictions.

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
