# Peer review of "A Neurosurgical Perspective on Brain Metastases from Renal Cell Carcinoma: Multi-Institutional, Retrospective Analysis"

_biomedicines, 2023, doi:10.3390/biomedicines11092485_

Round 1
Reviewer 1 Report
General comment
The manuscript entitled “A Neurosurgical Perspective on Brain Metastases from Renal Cell Carcinoma: Multi-institutional, Retrospective Analysis” aims to evaluate patients presenting brain metastases from RCC and treated neurosurgically. The topic is interesting and despite the limitations related to the small sample size, the paper could enrich the literature on the argument. Few corrections and additions are required before proceeding further with the peer-review.
ABSTRACT
25: the results should be more precise and provide objective findings. Remember that the abstract is the business card of your work and should be as much as possible representative of your effort.
INTRODUCTION
40: briefly report the recent advances in imaging. To this regard also see DOI: 10.3390/ijms24054615.
49: Considering the topic of your paper, provide epidemiological data on how many patients develop metastases and how many of them develop an head involvement. This does not need to be placed in this paragraph necessarily.
116: Albeit the introduction is well written and pleasurable to read, you should synthetize and highlight the topic of your work, clearly stating the aim of your study.
MATERIALS AND METHODS
150: Add inclusion and exclusion criteria, reporting the methodology of data retrievement. Be more precise as possible in this section. Additionally, specify if patients underwent to cytoreductive nephrectomy.
RESULTS
189: the entire subparagraph should be moved to the previous one. This section should start with the subparagraph 3.2
DISCUSSION
325: Also provide a brief overview on the role of cytoreductive nephrectomy. To this regard also see DOI: 10.3390/medicina59040767.
423: among the limitations consider also potential selection bias
minor grammar and typos errors
Author Response
ABSTRACT
25: The results should be more precise and provide objective findings. Remember that the abstract is the business card of your work and should be as much as possible representative of your effort. – We modified the results from the abstract section.
INTRODUCTION
40: Briefly report the recent advances in imaging. To this regard also see DOI: 10.3390/ijms24054615. – We added the requested information (lines 92-105).
49: Considering the topic of your paper, provide epidemiological data on how many patients develop metastases and how many of them develop a head involvement. This does not need to be placed in this paragraph necessarily. – We provided the requested information (lines 84-90).
116: Albeit the introduction is well written and pleasurable to read, you should synthesize and highlight the topic of your work, clearly stating the aim of your study. – We appreciate your great suggestion. In order to highlight the topic of our work we created subsection 1.4.3. A focus on Neurosurgery (lines 241-286).
MATERIALS AND METHODS
150: Add inclusion and exclusion criteria, reporting the methodology of data retrievement. Be more precise as possible in this section. Additionally, specify if patients underwent to cytoreductive nephrectomy. – We added the requested information.
RESULTS
189: The entire subparagraph should be moved to the previous one. This section should start with the subparagraph 3.2 – We believe that Reviewer 1 refers to lines 192-195 (Subparagraph “When comparing the distribution of lesion size….”), as line 189 represents the MRI of a BM RCC, and the next line (190) describes the Figure 1. We, therefore, moved this subparagraph to the previous one, and the entire section upward, in subparagraph 3.2.
DISCUSSION
325: Also provide a brief overview on the role of cytoreductive nephrectomy. To this regard also see DOI: 10.3390/medicina59040767. – Line 325 describes Figure 12, so we added a brief overview on role of cytoreductive nephrectomy at lines 228-240.
423: Among the limitations consider also potential selection bias – We discussed the matter (lines 605-610).
Comments on the Quality of English Language - minor grammar and typos errors – We revised the article and modified the identified mistakes, in order to provide a better quality of english language.
Reviewer 2 Report
Thank you for the opportunity to review this manuscript. Here, the authors provide a retrospective, multicentric study on 24 patients harboring cerebral renal cell carcinoma metastasis, 23 of which were neurosurgically resected.
It explores the role of neurosurgery in managing brain metastases from renal cell carcinoma (BM RCC), even in the era of targeted therapies. The authors analyze patient data from two neurosurgical departments in Romania, examining demographics, clinicopathological features, and treatment outcomes. They find better survival rates when patients receive multidisciplinary care involving nephrectomy, systemic therapy, and neurosurgery. The study underscores the importance of neurosurgery in treating symptomatic BM RCC and highlights the potential life-saving benefits of quick neurological symptom reversal.
The study is well-written and concise, and the data provided is thorough.
The manuscripts highlight the merits of multidisciplinary treatment, especially in a relatively small subgroup.
Author Response
We are extremely grateful for such feedback. Although we know that there is always room for improvement, and we are constantly trying to provide the best of our knowledge, we are encouraged by these types of reviews. Thank you and best regards.
Round 2
Reviewer 1 Report
The authors improved the manuscript accordingly to previous suggestion. No further comments from my side.
Minor typos and grammar checks